# Effect of Water on the Dynamic Tensile Mechanical Properties of Calcium Silicate Hydrate: Based on Molecular Dynamics Simulation

**DOI:** 10.3390/ma12172837

**Published:** 2019-09-03

**Authors:** Jikai Zhou, Yuanzhi Liang

**Affiliations:** College of Civil and Transportation Engineering, Hohai University, Nanjing 210098, China

**Keywords:** molecular dynamics simulation, calcium silicate hydrate, degree of saturation, strain rate, uniaxial tension

## Abstract

To study the effect of water on the dynamic mechanical properties of calcium silicate hydrate (C–S–H) at the atomic scale, the molecular dynamics simulations were performed in uniaxial tension with different strain rates for C–S–H with a degree of saturation from 0% to 100%. Our calculations demonstrate that the dynamic tensile mechanical properties of C–S–H decrease with increasing water content and increase with increasing strain rates. With an increase in the degree of saturation, the strain rate sensitivity of C–S–H tends to increase. According to Morse potential function, the tensile stress-strain relationship curves of C–S–H are decomposed and fitted, and the dynamic tensile constitutive relationship of C–S–H considering the effect of water content is proposed. This reveals the strain rate effect of the cementitious materials with different water content from molecular insights, and the dynamic constitutive relationship obtained in this paper is necessary to the modelling of cementitious materials at the meso-scale.

## 1. Introduction

Strong dynamic loadings, such as earthquake, explosion, impact, strong winds, and typhoons, etc. cause immeasurable losses for major cementitious structures. The improvement of dynamic mechanical properties is very important to prevent dynamic catastrophe of major projects, and many efforts have been dedicated to it [1,2,3,4,5,6].

Water content has a big effect on the cement-based materials [7]. Rossi et al. [8], Reinhardt et al. [9], Cadoni et al. [10], and Ross et al. [11] discovered that the dynamic strength of concrete increased for hydrated concrete under different strain rates, and dry concrete was insensitive to the strain rate. Zhou et al. [12] indicated for the totally dry concrete that the dynamic strength was 23% higher than that of the saturated concrete. Wu et al. [13] explored the effect of strain rates and water content on the mechanical behavior of concrete, and the results revealed that flexural strength increased with an increase in strain rates, while increasing water content would lower the flexural strength. Wang et al. [14] concluded that with the increase of strain rates, the dynamic strength of saturated and dry concretes increased, but the saturated structure had a higher strain rate sensitivity than dry structure in lateral restraint pressure.

C–S–H is the fundamental source of cement bond strength, which largely determines its unique mechanical properties [15]. Nevertheless, it is difficult to carry out relevant mechanical experiments at the nano/micro scale, so many researchers have investigated nano-micro properties by nanoscale simulation [16,17,18], and the molecular dynamics are of the most commonly used and effective methods for the studies at nanoscale. Tobermorite and Jennite minerals are regarded as the layered mineral analogue of C–S–H [19,20,21,22]. Many researchers have carried out experimental techniques to study the structural features of C–S–H, such as nuclear magnetic resonance imaging (NMR) [23] and small-angle neutron scattering (SANS) [23]. Pellenq et al. [24] proposed a bottom-up modeling method for building a C–S–H model, which was famous as a realistic model. Zhou et al. [25] developed a multi-scale simulation method based on molecular dynamics to obtain the mechanical properties of cement paste across scales, and the simulation results agreed with the experimental values. Lin et al. [26] found that the Hugoniot elastic limit of C–S–H gel was 7.5 GPa, and the it could cause different response regimes.

The number of water molecules in C–S–H gel are influenced by the surrounding environment [27]. Kalinichev et al. [28] demonstrated that the binding force of water on a solid surface was very strong. Pellenq et al. [24] compared the mechanical performances of C–S–H gels in dry and wet states and found that, during the loading process, the shear strength was weakened due to the obvious displacements of water molecules. Youssef et al. [29] explored the characteristics of water molecules in C–S–H gels and concluded that the H bonds were very strong, which constrained the movement of water molecules. Bonnaud et al. [30] investigated the cohesive force of C–S–H under different levels of humidity. The results showed that cohesive force depended on the interlayer space. Ji et al. [31] employed five water models to simulate the intrinsic properties of C–S–H gel. They found that the SPC model simulated the performance of C–S–H more accurately. Hou et al. [32,33] proved that the strength of bonds in the C–S–H gel decreased significantly due to the presence of water. 

Previous studies at the macro-scale have indicated that water content has an obvious effect on the dynamic mechanical behavior of cement-based materials, but few studies have been conducted on the influence mechanism. Since mechanical responses of materials are decided by their atomic structures and microstructures [34], it should reveal the effect of water content at the atomic scale. In this paper, C–S–H molecular structures with different degrees of saturation are proposed. Then the dynamic mechanical responses of C–S–H under different strain rates are studied by molecular dynamics. The influence mechanism of water content on the C–S–H is revealed at an atomic scale.

## 2. Materials and Methods 

In light of the method proposed by Pellenq [24], the atomic structure of C–S–H was constructed. The supercell (4a×3b×c) structure of anhydrous 11 Å Tobermorite (C/S = 1.0) was selected as the original structure of C–S–H. Then, we broke the silicate chains to meet the requirements of Q_n_ coefficient [35] and C/S [36], and anhydrous structure of C–S–H was obtained, as shown in Figure 1a. GCMC [37] simulation of the water adsorption was performed at 300 K, and SPC water model was adopted. The water content can be reflected by the degree of saturation, i.e., when the number of water molecules does not change during the adsorption process, the degree of saturation of C–S–H is considered to be 100%, and for other C–S–H structures with different water content, the degree of saturation is equal to the number of adsorbed water molecules divided by the number of adsorbed water molecules of the saturated C–S–H structures. The chemical formula of the saturated C–S–H is (CaO)_1.65_(SiO_2_)(H_2_O)_1.70_ and the density of saturated C–S–H is 2.43 g/cm^3^, which is quite close to the results in SANS [36]. The saturated C–S–H is in Figure 1b. In order to obtain stable statistical results, the C–S–H molecular structure was periodically duplicated in x, y, and z directions, and fully relaxed in the NPT ensemble. As for C–S–H, y is the direction of the silicate chain in C–S–H, z is the interlayer spacing direction, and x is the direction perpendicular to the yz plane of C–S–H. The size of the totally dry C–S–H molecular structure after equilibrium is x = 4.24 nm, y = 4.25 nm, z = 7.87 nm, and the size of the saturated C–S–H molecular structure is x = 4.45 nm, y = 4.51 nm, and z = 8.72 nm.

The MD modeling analyses are performed using the Lammps [38]. The C–S–H gels reach equilibrium states under constant pressure and temperature (NPT) for 100 ps, and the relaxation time step is 0.1 fs. The equilibrium configurations are achieved. ClayFF force field is used. The parameters in ClayFF force field are derived from experiments and ab initio calculations, which can well describe the mechanical behaviors of C–S–H and the thermodynamic properties of water molecules [39,40,41,42,43,44]. The force field parameters of ClayFF can be achieved from [39]. 

C–S–H interlayer spacing direction (z direction) is the weakest direction of strength [45], so z direction is chosen for study in this paper. To obtain the stress-strain relationship of C–S–H, the C–S–H with different degrees of saturation are subjected to uniaxial tensile loadings at constant strain rates of 0.0008 ps^−1^, 0.008 ps^−1^, 0.08 ps^−1^ and 0.8 ps^−1^ along the z direction, respectively. NPT ensembles are set in the whole simulation. The Verlet algorithm is chosen for integration of the motion equation. Periodic boundary conditions (PPP) are used in x, y, z dimensions in this paper. The simulation schemes are as follows: The eight different degrees of saturation are 0%, 7.59%, 14.17%, 20.38%, 39.62%, 61.07%, 79.51%, and 100%, respectively, and the strain rates are 0.0008 ps^−1^, 0.008 ps^−1^, 0.08 ps^−1^ and 0.8 ps^−1^. A total of 32 simulations were carried out.

## 3. Results and Discussion

### 3.1. Failure Pattern

Figure 2 shows the representative failure mode of the totally dry C–S–H and the saturated C–S–H under a high strain rate (0.8 ps^−1^) and low strain rate (0.0008 ps^−1^). It shows that (1) For the configurations of C–S–H in the failure stage, the destruction of dry C–S–H occurs in interlayer calcium layer, so the interlayer calcium layer is the weak layer of totally dry C–S–H and the saturated C–S–H. (2) Under the relatively high strain rate (0.8 ps^−1^), the damage of C–S–H is more uniform and there are several damage surfaces, which need more energy. However, under a relatively low strain rate (0.0008 ps^−1^), the damage of C–S–H is concentrated, and the ultimate damage surface is single, which requires less energy. 

### 3.2. Dynamic Tensile Stress-Strain Relationship Curves

The stress-strain relationship curves of C–S–H with a different degree of saturation under different strain rates are shown in Figure 3. It shows that there is a long yield platform in the stress-strain relationship curves under a relatively low strain rate (0.0008 ps^−1^, 0.008 ps^−1^), which proves that C–S–H exhibits good plasticity at a low strain rate. In addition, with an increase in the degree of saturation, the fracture strain of C–S–H tends to decrease, indicating that the increasing water content increases the brittleness of the C–S–H. However, under the relatively high strain rate (0.08 ps^−1^, 0.8 ps^−1^), the ultimate strength of C–S–H increases obviously, showing the same strain rate effect as that of the concrete structure at a macro scale, and with the increasing water content, there is no obvious yield platform.

### 3.3. Dynamic Tensile Strength and Young’s Modulus

The dynamic tensile strength and Young’s modulus are shown in Table 1 and Table 2. It shows that the dynamic peak tensile strength and Young’s modulus of C–S–H decrease with the increasing degree of saturation. Changes in the dynamic peak tensile strength with water content are consistent with experimental results of Wu [13]. Under relatively high strain rate (0.8 ps^−1^), the dynamic peak tensile strength of saturated C–S–H is about 29% lower than that of the totally dry C–S–H. However, under a relatively low strain rate at 0.0008 ps^−1^, the dynamic peak tensile strength of saturated C–S–H is about 36% lower than that of the totally dry C–S–H. This indicates that the degree of saturation has more influence on the dynamic strength when the strain rate is relatively low. Table 2 indicates when the strain rates increase from 0.0008 ps^−1^ to 0.8 ps^−1^, the Young’s modulus of saturated C–S–H decreases by 50%, 38%, 24%, and 20%, respectively, than that of the totally dry C–S–H, implying that the degree of saturation has a more obvious effect on it under a low strain rate.

### 3.4. Strain Rate Sensitivity of Dynamic Tensile Strength

To study the effect of degree of saturation on the dynamic strength of C–S–H, like the macroscopic dynamic experiments, the dynamic increase factor (DIF) is adopted, which is the ratio of the strength under high strain rate to the strength under low strain rate. The DIF is used to show the strain rate sensitivity of C–S–H with a different degree of saturation under dynamic loadings. At the atomic scale, the strength corresponding to the minimum strain rate (0.0008 ps^−1^) is chosen as the reference strength according to the strain rate of four different orders of magnitude applied.

As shown in Figure 4, the DIF increases nonlinearly with an increase in the strain rate, and it represents that when the strain rate is lower than 0.08 ps^−1^, the DIF value increases slowly. When the strain rate is higher than 0.08 ps^−1^, the DIF increases rapidly, indicating that the strain rate sensitivity of C–S–H dynamic strength is stronger with the increase of strain rate, and has the same rules at the macro scale [13]. When the strain rate is 0.8 ps^−1^, the DIF value of saturated C–S–H is higher than that of totally dry C–S–H. The higher the strain rate, the larger the range of DIF values of C–S–H with different degrees of saturation.

### 3.5. Influence Mechanism of Water Content on Tensile Mechanical Properties

#### 3.5.1. Volume Deformation of C–S–H

C–S–H molecular structures in the equilibrium state gradually change to a saturated state, owing to an increase in water content. Eight C–S–H molecular structures were obtained with a degree of saturation of 0%, 7.59%, 14.17%, 20.38%, 39.62%, 61.07%, 79.51%, and 100%. Therefore, when water molecules enter the C–S–H structures, the initial deformation of C–S–H structures occurs relative to the totally dry C–S–H structure in the z direction.

The formula of Δz varying with degree of saturation can be obtained by Equation (1).

(1)$$\Delta z=0.067+0.0083S_{r}\;R^{2}=0.98$$
where Δz is the deformation of C–S–H in the z direction; $S_{r}$ is the degree of saturation; $R^{2}$ is the correlation coefficient

#### 3.5.2. Influence Mechanism of Water Molecules 

When water molecules enter the C–S–H structure cell, the interaction between atoms in C–S–H changes. The left schematic diagram in Figure 5 is the totally dry C–S–H. And after a single water molecule enters the C–S–H interlayer, as shown in the middle schematic diagram in Figure 5, it would have an effect on the interaction between atoms near the water molecule. On one hand, because the distance between atoms around the water molecule increases, the interaction is weakened. On the other hand, the interaction between the water molecule and the ambient atoms in the C–S–H matrix also exists, which conforms to the multi-body potential interaction rule, as shown in the middle schematic diagram. It is observed in the right schematic diagram that the volume of C–S–H structure cell continues to expand as the number of adsorbed water molecules increase.

The C–S–H structure after water absorption can be regarded as a composite of water molecules and C–S–H matrix (without water molecule). In Table 3, the initial deformation of the C–S–H composite in an equilibrium state has occurred due to the volume expansion of the C–S–H after water absorption. Therefore, the distance between atoms interaction in C–S–H composite is larger than that in an absolutely dry state, which weakens the tensile strength of C–S–H matrix. Meanwhile, water molecules also interact with surrounding C–S–H matrix atoms, which is beneficial to tensile strength of C–S–H composite, but relatively weak. Therefore, after water adsorption, the tensile strength and Young’s modulus are weaker than those of absolutely dry C–S–H, and decrease with the increase of water content.

According to the analysis above, the tensile interaction of C–S–H composite after water absorption can be decomposed into two parts. One part is the interaction between atoms of C–S–H matrix. The other part is the interaction between water molecules and the surrounding C–S–H matrix atoms. Based on Morse potential function, the C–S–H tensile stress-strain relationship curves with different strain rates and degree of saturation were decomposed and fitted, and from which a C–S–H dynamic tensile stress-strain relationship formula considering the effect of degree of saturation was proposed.

In order to analyze the C–S–H tensile stress-strain relationship curve, the following basic assumptions are proposed:

(1) The interaction potential between water molecules and C–S–H matrix atoms conforms to Morse potential function law;

(2) The residual peak tensile strength ($f_{R,P,S_{r}}$) of C–S–H matrix weakened by water molecules is equal to the peak tensile strength ($f_{P,d}$) of totally dry C–S–H subtracting the initial stress($\sigma_{S_{r}}$) due to initial strain in the tensile stress-strain relationship curve of totally dry C–S–H after water absorption;

(3) The peak tensile strain of C–S–H matrix decomposed by C–S–H composite with different degree of saturation is consistent with that of a totally dry C–S–H matrix;

(4) The stress of C–S–H matrix with different degree of saturation is obtained by multiplying the stress of absolutely dry C–S–H matrix by the reduction coefficient (η), calculated by the following formula:(2)$$\eta =1-\frac{\sigma_{S_{r}}}{f_{P,d}}$$

Based on assumptions, the stress-strain relationship curves of C–S–H matrix with different degree of saturation are obtained. In order to obtain the stress values of the interaction between water and C–S–H matrix, subtract the stress of C–S–H matrix with the same degree of saturation from the stress of C–S–H composite at the same strain level. Thus, the stress-strain relationship curves of C–S–H matrix with a different degree of saturation and the stress-strain relationship curves of the interaction between water molecules and C–S–H matrix can be obtained.

Morse potential function [46] is a potential function of classical interaction between atoms, as shown in Equation (3).

(3)$$U_{Morse}(r)=D\left({\left(1-\exp\left(-\beta \left(r-r_{0}\right)\right)\right)}^{2}-1\right)$$
where D and $r_{0}$ indicate potential function parameters, D is potential well, r is the distance between two atoms, $r_{0}$ is the distance between two atoms at the lowest potential energy, β is shape parameter of potential function curve.

Based on Morse potential function, the tensile stress-strain relationship of C–S–H with different water content is proposed.

(4)$$\sigma_{s}=f_{P,S_{r}}(1-{\left(1-\exp(-\beta_{S_{r}}(\varepsilon_{s}-\varepsilon_{P}))\right)}^{2})$$
where $\sigma_{s}$ is stress, in GPa; $\varepsilon_{s}$ is tensile strain; $f_{P,S_{r}}$ is peak tensile strength with different degree of saturation, in GPa; $\varepsilon_{P}$ is peak tensile strain; $\beta_{S_{r}}$ is shape parameters of stress-strain relationship curve with different degree of saturation.

According to the influence mechanism of water molecules on the interaction between C–S–H atoms after water adsorption, the stress-strain relationship curves of C–S–H with different water content under uniaxial tension with the strain rate of 0.8 ps^−1^ were analyzed. The total stress-strain relationship curves in Figure 6a of C–S–H under uniaxial tension with strain rate of 0.8 ps^−1^ and the stress-strain relationship curves Figure 6b of decomposed C–S–H matrix and the stress-strain relationship curves Figure 6c of interaction between water molecule and C–S–H matrix were fitted by Equation (4), and fitted stress-strain relationship curves are also shown as follows.

The minimum correlation coefficient of the fitted results of the stress-strain curves in Figure 6 is 0.97. It can be seen that Equation (4) fits well and can accurately express the constitutive relation of C–S–H with different water content. In Figure 6, the fitted parameters are obtained in Table 4.

According to the analysis above, Equation (4) can well fit the stress-strain relationship curves. The stress-strain relationship curves of the decomposed C–S–H matrix and the stress-strain relationship curves of the interaction between water molecules and C–S–H matrix are also fitted well by the Equation (4). 

Therefore, in this paper, MD and GCMC simulation methods are adopted and based on the Morse potential function, the tensile stress-strain relationship of C–S–H with different water content is proposed. From the fitted results, it can be found that it is reasonable to reveal the influence mechanism of water molecules on the tensile mechanical properties of C–S–H at atomic scale, and the calculation assumptions and formulas are applicable.

### 3.6. Dynamic Tensile Constitutive Relationship

For constructing the dynamic tensile constitutive relationship of C–S–H with different degree of saturation, the tensile stress-strain relationships of C–S–H under different strain rates are analyzed. Without losing generality, based on Morse potential function, the dynamic tensile constitutive relationship of C–S–H with different degree of saturation under different strain rates is proposed.

(5)$$\sigma_{S_{r}}^{d}=f_{P,S_{r}}^{d}(1-{\left(1-\exp(-\beta_{{}^{S_{r}}}^{d}(\varepsilon_{{}^{S_{r}}}^{d}-\varepsilon_{P,S_{r}}^{d}))\right)}^{2})$$
where $\sigma_{S_{r}}^{d}$ is dynamic tensile stress of C–S–H with different degree of saturation, in GPa; $\varepsilon_{{}^{S_{r}}}^{d}$ is C–S–H tensile strain with different degree of saturation; $f_{P,S_{r}}^{d}$ is dynamic peak tensile strength of C–S–H with different degree of saturation, in GPa; $\varepsilon_{P,S_{r}}^{d}$ is dynamic peak tensile strain with different degree of saturation; $\beta_{{}^{S_{r}}}^{d}$ is the shape parameter of dynamic stress-strain curve with different degree of saturation.

According to the above mechanism analysis of interaction between C–S–H atoms, the tensile stress-strain relationship curves of C–S–H and the decomposed tensile stress-strain relationship curves at strain rates of 0.8 ps^−1^, 0.08 ps^−1^, 0.008 ps^−1^ and 0.0008 ps^−1^ were fitted respectively, the fitted parameters $f_{P,S_{r}}^{d}$ are shown in Figure 7.

According to Figure 7, under different strain rates, the peak tensile strength of C–S–H with different degree of saturation is shown in Equation (6).

(6)$$f_{P,S_{r}}^{d}=f_{P,S_{0}}^{d}\exp(-\alpha_{d}S_{r})$$
where $f_{P,S_{0}}^{d}$ is the dynamic peak tensile strength of totally dry C–S–H, in GPa; $\alpha_{d}$ is the coefficient related to strain rates.

Fitted values of $f_{P,S_{0}}^{d}$, $\alpha_{d}$ are shown in Table 5:

In Table 5, the relationships of $f_{P,S_{0}}^{d}$, $\alpha_{d}$ and strain rates $\overset{\cdot }{\varepsilon }$ are analyzed, as shown in Equations (7) and (8).

(7)$$f_{P,S_{0}}^{d}=5.24\exp{\left[3.53\cdot \mathrm{lg}\left(\overset{\cdot }{\varepsilon }\right)\right]}_{}\;R^{2}=0.98$$

(8)$$\alpha_{d}=3.76-0.0006\cdot \mathrm{lg}\left(\overset{\cdot }{\varepsilon }\right)\;R^{2}=0.94$$

The relationship between peak tensile strength $f_{P,S_{r}}^{d}$ of C–S–H with degree of saturation and strain rates is proposed, as shown in Equation (9):(9)$$f_{P,S_{r}}^{d}=\left[5.24\exp\left[3.53\cdot \mathrm{lg}\left(\overset{\cdot }{\varepsilon }\right)\right]\right]\cdot \exp\left[\left(-3.76+0.0006\cdot \mathrm{lg}\left(\overset{\cdot }{\varepsilon }\right)\right)\cdot S_{r}\right]$$

According to the above fitted results, it can be seen that the variation of $\beta_{S_{r}}$ and $\varepsilon_{P,S_{r}}$ with strain rates and degree of saturation is not large, so the average value of $\beta_{S_{r}}$ and $\varepsilon_{P,S_{r}}$ are adopted respectively as 4.79 and 0.148. Therefore, the dynamic stress-strain relationship of C–S–H with different water content under different strain rates can be obtained by combining the proposed constitutive Equations (5) and (9), as shown in Equation (10).

(10)$$\sigma_{S_{r}}^{d}=\left\{\left[5.24\exp\left[3.53\cdot \mathrm{lg}\left(\overset{\cdot }{\varepsilon }\right)\right]\right]\cdot \exp\left[\left(-3.76+0.0006\cdot \mathrm{lg}\left(\overset{\cdot }{\varepsilon }\right)\right)\cdot S_{r}\right]\right\}\left[\left(1-{\left(1-\exp(-4.79(\varepsilon -0.148))\right)}^{2}\right)\right]$$
where $\sigma_{S_{r}}^{d}$ is the dynamic stress of C–S–H with different saturation, in GPa, ε is tensile strain.

## 4. Conclusions

In this study, the effect of the degree of saturation (0%, 7.59%, 14.17%, 20.38%, 39.62%, 61.07%, 79.51%, and 100%) on the dynamic tensile properties of C–S–H under different strain rates (0.0008 ps^−1^, 0.008 ps^−1^, 0.08 ps^−1^, 0.8 ps^−1^) have been revealed.

(1) Under the same strain rate, the dynamic mechanical behaviors of C–S–H composite, such as the tensile strength and Young’s modulus, decrease with the increasing water content;

(2) The effect of water molecules on the tensile behaviors of C–S–H is revealed at the atomic scale. The interaction between the atoms in C–S–H matrix around water molecules is weakened when water molecules enter the C–S–H structure, which results in the increased distance between the atoms in C–S–H matrix. Meanwhile, water molecules interact with the surrounding atoms in C–S–H matrix, which is beneficial to the tensile strength, but relatively weak;

(3) The tensile interaction of C–S–H after water absorption can be decomposed into two parts: one part is the interaction of atoms in the C–S–H matrix weakened by water molecules, the other is the interaction between water molecules and the surrounding C–S–H matrix atoms; The stress-strain relationship curves of C–S–H composites with different strain rates and degree of saturation are decomposed and fitted based on Morse potential function, the dynamic tension stress-strain relationship of C–S–H considering the effect of water content is proposed;

(4) The dynamic peak tensile strength and Young’s modulus of C–S–H with the same degree of saturation increase with the increase of strain rate, showing obvious strain rate effect. With an increase in water content, the strain rate sensitivity of C–S–H increases.

Our findings provide a new understanding of the effect of water molecules on the dynamic mechanical properties of cementitious materials at the molecular level. The dynamic constitutive relationship in this paper for C–S–H nanostructures is fundamental for a multi-scale study on dynamic mechanical properties of cement-based materials. 

## Figures and Tables

**Figure 1 materials-12-02837-f001:**
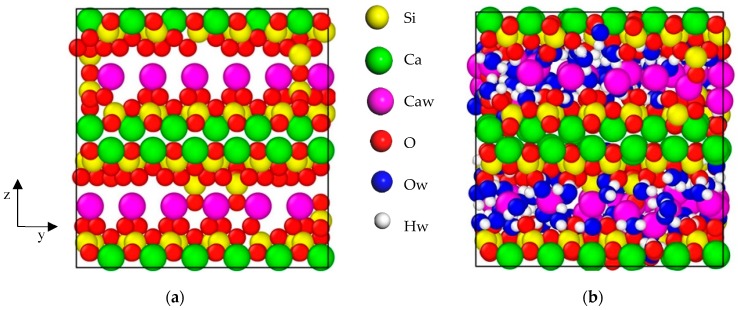
(**a**) Dry C–S–H (**b**) saturated C–S–H.Simulation box size: (**a**) a = 2.23 nm, b = 2.22 nm, c = 2.28 nm; (**b**) a = 2.20 nm, b = 2.18 nm, c = 2.24 nm. Yellow ball, green ball, pink ball, red ball, blue ball and white ball represent the silicon atom (Si), layered calcium atom (Ca), interlayer calcium (Caw), oxygen atom (O) in the silicate chain, oxygen atom (Ow) in water and hydrogen atom (Hw) in water respectively.

**Figure 2 materials-12-02837-f002:**
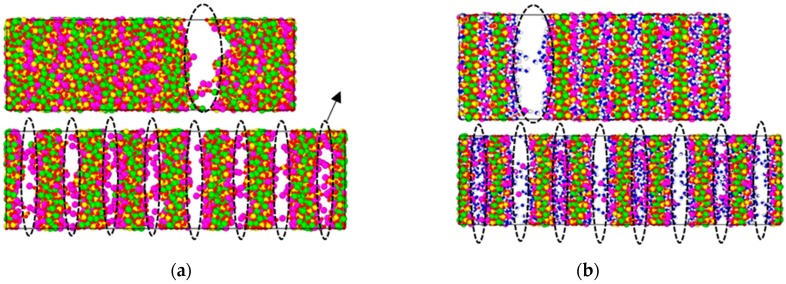
Under tensile loading with strain rate at 0.8 ps^−1^(below) and 0.0008 ps^−1^(above), the failure mode of (**a**) totally dry C–S–H (**b**) saturated C–S–H.

**Figure 3 materials-12-02837-f003:**
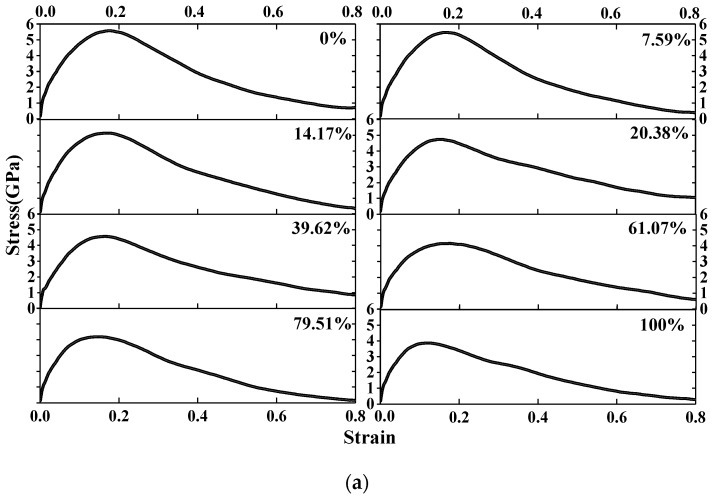
Stress-strain relationship curves of C–S–H with different degrees of saturation under different strain rates. **(a)** 0.8 ps^−1^; **(b)** 0.08 ps^−1^; **(c)** 0.008 ps^−1^; **(d)** 0.0008 ps^−1.^

**Figure 4 materials-12-02837-f004:**
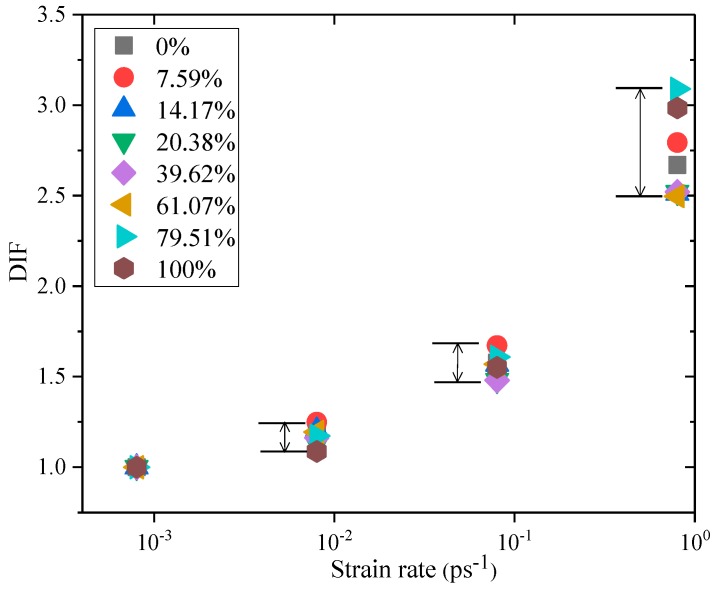
Relationship between dynamic increase factor of C–S–H and strain rate at different degree of saturation.

**Figure 5 materials-12-02837-f005:**
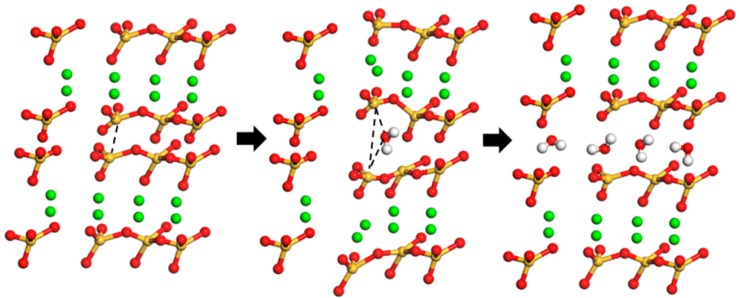
Diagram of interlayer interaction after water molecules entering C–S–H structure. Yellow ball, green ball, red ball connected with yellow ball, red ball connected with white ball and white ball represent the silicon atom (Si), layered calcium atom (Ca), oxygen atom (O) in the silicate chain, oxygen atom (Ow) in water and hydrogen atom (Hw) in water respectively.

**Figure 6 materials-12-02837-f006:**
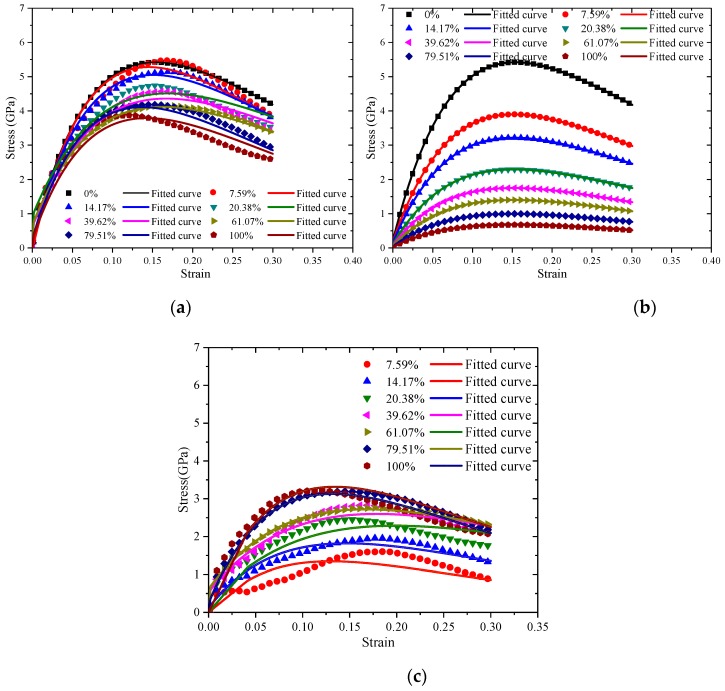
Under the strain rate at 0.8 ps^−1^, decomposition and fitting of stress-strain relationship curves for the effect of water on the mechanical properties: (**a**) Total effect of water; (**b**) Effect of water on C–S–H matrix; (**c**) Interaction between water molecules and C–S–H matrix.

**Figure 7 materials-12-02837-f007:**
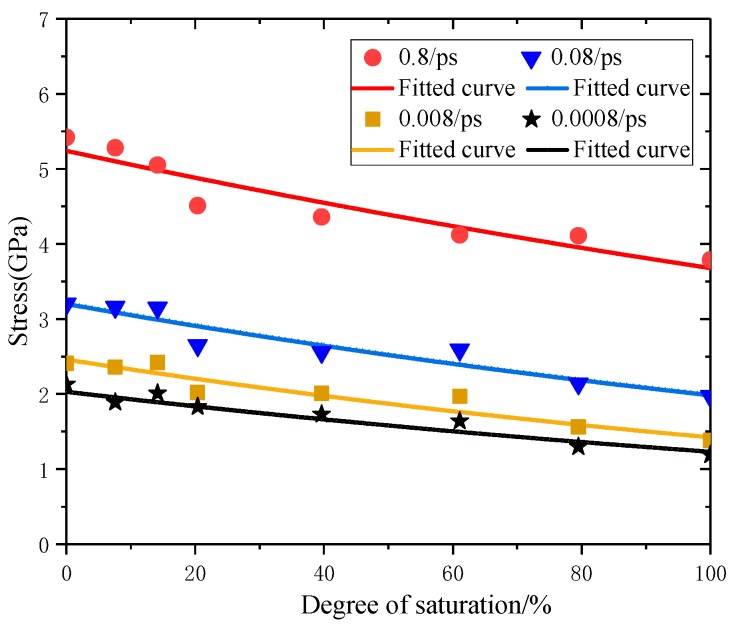
Fitted parameters $f_{P,S_{r}}^{d}$ under different strain rates.

**Table 1 materials-12-02837-t001:** Tensile strength of C–S–H with different degree of saturation under different strain rates (GPa).

Strain Rate(ps^−1^)	Degree of Saturation							
	0%	7.59%	14.17%	20.38%	39.62%	61.07%	79.51%	100%
0.0008	2.03	1.89	2.01	1.79	1.73	1.65	1.33	1.27
0.008	2.41	2.36	2.42	2.02	2.01	1.97	1.56	1.38
0.08	3.20	3.16	3.15	2.65	2.56	2.59	2.14	1.97
0.8	5.57	5.47	5.14	4.51	4.36	4.12	4.11	3.79

**Table 2 materials-12-02837-t002:** Young’s modulus of C–S–H with different the degree of saturation under different strain rates (GPa).

Strain Rate(ps^−1^)	Degree of Saturation							
	0%	7.59%	14.17%	20.38%	39.62%	61.07%	79.51%	100%
0.0008	40.11	38.33	37.10	36.10	32.02	28.17	26.14	20.01
0.008	42.05	41.01	38.52	38.12	35.06	33.16	31.20	26.23
0.08	54.08	52.12	51.05	50.04	47.09	46.02	44.13	41.08
0.8	72.16	70.08	68.22	67.04	66.14	64.11	61.18	58.03

**Table 3 materials-12-02837-t003:** Deformation of C–S–H in the z direction (interlayer spacing direction) with different degree of saturation.

	Degree of Saturation							
	0%	7.59%	14.17%	20.38%	39.62%	61.07%	79.51%	100%
Δz/nm	0	0.070	0.197	0.299	0.496	0.589	0.724	0.85

Δz, deformation in *z* direction.

**Table 4 materials-12-02837-t004:** The parameters fitted by Equation (4) in Figure 6.

Degree of Saturation/Fitted Parameters	$f_{P,S_{r}}\;(a)\;(\mathrm{GPa})$	$\beta_{S_{r}}\;(a)$	$\varepsilon_{P}\;(a)$	$f_{P,S_{r}}\;(b)\;(\mathrm{GPa})$	$\beta_{S_{r}}\;(b)$	$\varepsilon_{P}\;(b)$	$f_{P,S_{r}}\;(c)\;(\mathrm{GPa})$	$\beta_{S_{r}}\;(c)$	$\varepsilon_{P}\;(c)$
0%	5.42	4.44	0.154	5.42	4.44	0.154	0	5.79	0.128
7.59%	5.28	4.80	0.145	3.90	4.44	0.154	1.38	4.60	0.147
14.17%	5.05	4.52	0.151	3.22	4.44	0.154	1.82	3.07	0.190
20.38%	4.51	3.77	0.169	2.29	4.44	0.154	2.21	3.559	0.179
39.62%	4.36	3.93	0.167	1.75	4.44	0.154	2.60	3.78	0.167
61.07%	4.12	4.11	0.162	1.40	4.44	0.154	2.72	4.1	0.132
79.51%	4.11	4.74	0.140	0.996	4.44	0.154	3.24	4.5	0.130
100%	3.97	4.73	0.134	0.675	4.44	0.154	3.13	5.79	0.128

**Table 5 materials-12-02837-t005:** The parameters fitted by Equation (6) in Figure 7.

Strain Rate /ps^−1^	$f_{P,S_{0}}^{d}/\mathrm{GPa}$	$\alpha_{d}$
0.8	5.24	0.0035
0.08	3.20	0.0047
0.008	2.47	0.0056
0.0008	2.03	0.0050

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
