# Peer review of "Effect of Water on the Dynamic Tensile Mechanical Properties of Calcium Silicate Hydrate: Based on Molecular Dynamics Simulation"

_materials, 2019, doi:10.3390/ma12172837_

Round 1
Reviewer 1 Report
33-36 Please explain sentence: dynamic compressive strength of cement mortar increased with an increase in water content and dynamic compressive strength of the saturated concrete is lower than totally dry
93 4a( a italics)
107 Please explain directions x, y, z
110 What are a=2.23nm, b=2.22nm, c=2.28nm dimensions? Please mark in figure 1
114 Lammps or Limpton?
143 totally not toally
145 totally not toally
149 yield platform ? maybe curve?
151-153 Sentence: "in addition....." is correct for strain rate in figure 4 a, b, c. For d we can't state clearly
155 yield platform ? maybe curve?
162 about 29%
163 about 36% this same in 166
198 Please explain directions z
199 What is the dimension of the model in Figure 6?
205 Formula (1) please explain △z, Sr, and R2
217 Please explain colours like in figure 1. (keep compatibility)
243 Please explain σsr
245 - 246 What it comes from?
251 - 254 Incomprehensible sentence, are You sure that in line 252 should be "by"?
258 Formula (3) please explain r
263 σs is tensile stress, εs what type of the strain
266 Adsorption or absorption?
271 Line in figure 8 (c) should have this same typ what in a and b for respectively degrees of saturation
281 Please check if the values in the table are consistent with the drawing, in this scale line i think that for saturated ) should be more than 5.5
282 "The As shown" ? i don't know what it means. fp,sr is described peak tensile strength ( this same is in line 263, but with different symbols), i think that should be i. e. peak tensile strength for adequate saturation.
283,289 Forumla (5) - I think it is a big mistake and it should be 5.42 not 5.24. Please check if the calculations are correct !!!
287 Forumla (6) - I think it is a big mistake and it should be 5.42 not 5.24. Please check if the calculations are correct !!!
293 One more time described peak tensile strength but symbols are different than in line 263.
314 - 316 All symbols should have a description: "with different degree of saturation" or " i. e. for "adequate saturation"
320 I think that should be fdP,Sr not